# Cefazolin Might Be Adequate for Perioperative Antibiotic Prophylaxis in Intra-Abdominal Infections without Sepsis: A Quality Improvement Study

**DOI:** 10.3390/antibiotics11040501

**Published:** 2022-04-10

**Authors:** Güzin Surat, Pascal Meyer-Sautter, Jan Rüsch, Johannes Braun-Feldweg, Christian Karl Markus, Christoph-Thomas Germer, Johan Friso Lock

**Affiliations:** 1Unit for Infection Control and Antimicrobial Stewardship, University Hospital of Würzburg, 97080 Würzburg, Germany; 2Department of General, Visceral, Transplant, Vascular and Pediatric Surgery, University Hospital of Würzburg, 97080 Würzburg, Germany; meyer-sautter.p@gmx.de (P.M.-S.); jan-ruesch@t-online.de (J.R.); johannes.braun-feldweg@gmx.de (J.B.-F.); germer_c@ukw.de (C.-T.G.); lock_j@ukw.de (J.F.L.); 3Department of Anesthesiology, Intensive Care, Emergency and Pain Medicine, University Hospital of Würzburg, 97080 Würzburg, Germany; markus_c@ukw.de

**Keywords:** antimicrobial stewardship, antibiotic prescribing quality, low-risk intra-abdominal infections, perioperative antibiotic prophylaxis

## Abstract

Background: The adequate choice of perioperative antibiotic prophylaxis (PAP) could influence the risk of surgical site infections (SSIs) in general surgery. A new local PAP guideline was implemented in May 2017 and set the first-generation cefazolin (CFZ) instead the second-generation cefuroxime (CXM) as the new standard prophylactic antibiotic. The aim of this study was to compare the risk of SSIs after this implementation in intra-abdominal infections (IAIs) without sepsis. Methods: We performed a single center-quality improvement study at a 1500 bed sized university hospital in Germany analyzing patients after emergency surgery during 2016 to 2019 (*n* = 985), of which patients receiving CXM or CFZ were selected (*n* = 587). Propensity score matching was performed to ensure a comparable risk of SSIs in both groups. None-inferiority margin for SSIs was defined as 8% vs. 4%. Results: Two matched cohorts with respectively 196 patients were compared. The rate of SSIs was higher in the CFZ group (7.1% vs. 3.6%, *p* = 0.117) below the non-inferiority margin. The rate of other postoperative infections was significantly higher in the CFZ group (2.0% vs. 8.7%, *p* = 0.004). No other differences including postoperative morbidity, mortality or length-of-stay were observed. Conclusion: Perioperative antibiotic prophylaxis might be safely maintained by CFZ even in the treatment of intra-abdominal infections.

## 1. Introduction

Surgical site infections (SSIs) are the most dreaded infectious complications in patients undergoing intra-abdominal surgery and account for the third common cause of nosocomial infections [1,2,3]. Appropriately administered perioperative prophylaxis (PAP) remains, embedded in tailored perioperative infection prevention bundles, one of the most indispensable measures in the prevention of SSIs [4,5,6,7]. Appropriateness of PAP defines timing, dosage, duration and choice of antimicrobials and given the fact that up to 15% of prescribed antibiotic agents for inpatients are for PAP and again approximately 45% are categorized inappropriate, fueling antimicrobial resistance and boosting healthcare costs, interventions on optimizing the prescription behaviour on the use for surgical antibiotic prophylaxis are one of the main targets of antimicrobial stewardship programs (AMS) [2,8,9,10]. In accordance with national and international guidelines and in order to fullfill the German Act on the Prevention and Control of Infectious Diseases (Infektionsschutzgesetz § 23) the university hospital of Würzburg (UKW) established an in-house antimicrobial stewardship program (ASP) introducing antibiotic prescription and treatment protocols along with hospital wide antibiotic wards rounds [11,12,13]. The first developed hospital guideline on PAP marked a change in the drug selection: in keeping with the current effective clinical practice guidelines for antimicrobial prophylaxis the 2nd generation cephalosporin cefuroxime (CXM) was substituted for the 1st generation cephalosporin cefazolin (CFZ) [7]. Appropriateness including adherence to treatment protocols represent important quality improvement indicators for procedure-associated infections, so the aim of this study was first, to monitor the compliance and second, to analyze the impact and concordance on SSIs and other postoperative infectious complications in non-elective uncomplicated and complicated intra-abdominal infections (IAIs) comparing both named agents [14,15]. Possible alterations on the choice for postoperative treatment (PAT) antibiotic regimens were not less of interest and will be discussed shortly too.

## 2. Methods

This quality improvement study entails a period of 4 years (2016–2019) and was conducted retrospectively in a 1500 bed tertiary hospital in Germany, with an in-hospital ASP officially launched in 2015. The backbone of the in-house AMS team consists of infection control physicians, microbiologists, pharmacists and infectious disease (ID) consultants with an ID physician responsible for the leadership. The fist prequel of this project included 776 patients during 2016–2018 and focused on the impact of antimicrobial stewardship interventions on surgical antibiotic prescription behavior of surgical IAIs, especially postoperative antibiotic use and the appropriateness of indication. The mentioned analysis revealed a significant reduction of total days of antibiotic therapy and fewer patients receiving postoperative antibiotic therapy altogether [16]. The intention of this subsequent analysis was to assess the impact of antimicrobial stewardship implementations on PAP and—if required—PAT.

### 2.1. Study Design

A structured local PAP guideline was implemented in May 2017 and set CFZ instead CXM as the new standard prophylactic antibiotic. All patients undergoing emergency surgery for IAIs during 2016–2019 were analyzed and characteristics and outcomes of patients receiving CFZ vs. CXM were differentiated. All data were retrieved from the hospital information system and transferred in a pseudonymous database with multiple variables containing baseline patient characteristics, pre-, peri- and postoperative antibiotic therapy (ABT), surgical therapy, and postoperative 30-day outcome. Adequate surgical source control was defined as prompt and adequate control over the abdominal source of infection including all measures undertaken to remove the source of infection, decrease the bacterial contamination and correct anatomic derangements to recover normal physiologic function. Postoperative complications were graded according to Clavien-Dindo classification [17]. Clavien-Dindo grade I-II complications were appraised as no severe complications, whereas Clavien-Dindo grade IIIa-V complications were appraised as severe complications. The follow-up was limited to 30 days.

### 2.2. Patients

All patients ≥ 18 years requiring emergency abdominal surgery for IAIs during 1 January 2016 and 31 December 2019 were screened for the following selection criteria: firstly, all patients with acute pancreatitis, acute mesenteric ischemia, acute leukemia, end-stage malignant disease in palliative care, ASA score > IV, and extra-abdominal infectious focus requiring antimicrobial therapy before and after surgery were excluded. From the remaining patients (*n* = 985), only those who received PAP or PAP and PAT with CFZ or CXM and without sepsis were selected (*n* = 587; 59.6%).

### 2.3. Statistical Analysis

All statistical analyses were performed using IBM SPSS Statistics, version 27 (International Business Machines Corporation, Armonk, NY, USA). Descriptive data were reported as means with standard deviation, unless otherwise noted. Groups were compared using the Chi-square, Fisher’s exact Test or Mann–Whitney U test according to the data scale and distribution. The level of statistical significance was 0.05 (two-sided). The primary endpoint was defined as the incidence of SSIs according to the Centers for Disease Control and Prevention (CDC) criteria [18]. The secondary endpoints included postoperative infections other than SSIs, the rate of patients requiring postoperative escalation in antibiotic therapy, the duration of PAT and length-of-stay (LOS).

Assuming an overall SSIs rate of 4% in both groups with a doubled non-inferiority level of 8% (as an acceptable upper limit for SSI after colorectal surgery), we intended to analyze a minimum of 312 patients with a power of 95% at a two-sided α of 0.05 to show non-inferiority of incidence of SSIs. The overall comparison of the CFZ and the CXM group revealed significant differences in the patient characteristics. Consequently propensity score (PS) matching was used to reduce the bias from these potential confounding variables. Based on the available literature, we chose five variables that predict the risk of postoperative SSIs: the focus of IAIs, community- vs. hospital-acquired infections, the patients’ sex, the individual comorbidity scaled by the Charlson Comorbidity Index and the necessity of PAT. These variables were entered a logistic regression analysis to calculate PS. A 1:1 matching ratio (CXM vs. CFZ) was chosen. A caliper of width equal to 0.2 of the standard deviation (SD) of the logit of the PS was used. The PS matching yielded two groups, each containing 196 patients.

## 3. Results

### 3.1. Changes in Perioperative Antibiotic Prophylaxis and Postoperative Treatment

Until May 2017 CXM had been the local standard agent for PAP (in combination with the anti-anaerobic antibiotic metronidazole if required). Equally 100% of patients in 2016 received CXM, not one patient CFZ. Vice versa the amount of patients receiving CFZ for PAP reached then nearly 100% in 2019. The PAP was applied within 30 minutes prior skin incision by intravenous push infusion.

As expected the new PAP guideline effected the PAT too: CXM was completely replaced by CFZ. Interestingly, the percentage of patients receiving PAT slightly decreased from 33% to 26%, but without any statistical significance (Table 1).

### 3.2. Patient Characteristics and Intraoperative Findings

Detailed description of all patients before PS matching are provided in Table 2. Both analyzed cohorts consisted of patients with a mean age of <50 years with no or low comorbidity in >65%, the incidence of MDR was <5%, and >90% of the IAIs was community-acquired, mostly acute appendicits. Surgical therapy was performed by laparoscopy in >60% of cases and >70% of the patients were successfully managed on general wards. Adequate source control was achieved in >99% of the patients.

Basic characteristics such as age, sex, BMI, risk scores (Charlson comorbidity index and ASA score) showed no relevant variation between the groups. The rate of severe liver or kidney disease, immunosuppression or malignancy at the time of surgery was low in both groups. However, a significant difference concerning the surgical focus within the groups was observed: the CFZ group displays a higher percentage of obstructive ileus (19.2% vs. 3.1%) while CXM group displays a higher percentage of acute cholecystitis (15.8% vs. 33.2%).

### 3.3. Characteristics after Propensity Score Matching

During PS matching 195 patients were excluded from analysis mostly due to the 1:1 matching ratio. However, the matched cohorts were even more focused on younger patients with fewer comorbidities, community acquired IAIs and laparoscopic therapy. Interestingly the PS matching changed the sex ratio within the CFZ groups, while all other variables became homogenous between both groups (Table 3).

### 3.4. Antibiotic Therapy and Microbiology Findings

Details on antibiotic therapy and microbial findings are provided in Table 4. No significant differences were seen in the management of antibiotic therapies. CFZ was administered in median one day shorter that CXM (5 vs. 6 days). Thirteen out of 273 patients (4.7%) who only received single-shot PAP during surgery received a later postoperative antibiotic therapy. Out of these, three patients developed SSI and 3 patients developed other infection. Later PAT was indicated due to clinical deterioration in 11 patients after 2–5 days, while 2 patients received later PAT due to the resistogram. The most common reason for a change in antibiotic therapy in patients with PAT was clinical deterioration (64% CXM vs. 72% CFZ). Switches occurred after a median of 3 (2–5) days. The switches were largely associated with elevated CRP (median 11 vs. 24; *p* < 0.001). In 74% of these cases, no postoperative infection was diagnosed nor any focus intervention was required.

Bacterial species detected in intraoperative swabs were also comparable containing a broad variety of gram positive, negative and anaerobic stains. The most frequent bacteria found in positive stains was *E. coli* in >50% of both groups. MDR stains were found in only 1% of patients in both groups.

### 3.5. Postoperative Outcome

Details on postoperative outcome are provided in Table 5. The incidence of SSIs was two-fold higher in the CFZ group without reaching statistical significance. Out of these, the incidence of SSIs was higher in those patients requiring PAT (8.2% with CXM, 13.8% with CFZ) in comparison to single-shot PAP (1.5% with CXM, 4.3% with CFZ). In contrast, the incidence of other postoperative infections was significantly higher within the CFZ group, mainly for urinary tract infections. In accordance, the incidence of other infections was also higher in patients requiring PAT (1.6% with CXM, 13.8% with CFZ; *p* = 0.013) in comparison to single-shot PAP (2.2% with CXM, 6.5% with CFZ).

Furthermore, no changes of postoperative morbidity and mortality, as well as, LOS were observed.

## 4. Discussion

The intention of this quality improvement study was to analyze performance measures for infectious complications like SSIs and other postoperative infections after emergency surgery in non-septic patients with intra-abdominal infections in the context of in-house AMS implementations. CXM for PAP was compared with CFZ for the same purpose. Our results demonstrate that in fact the rate of SSIs was higher in the CFZ group, but without any statistical significance and that no differences including postoperative morbidity, mortality or length-of-stay were observed. On the contrary to the prequels published by Surat and al. this study focused mainly on the implications of ‘de-escalating’ the by then commonly for surgical prophylaxis administered antibiotic CXM, a 2nd generation cephalosporin to CFZ, a 1st generation cephalosporin [16,20]. First-, second-, and third-generation cephalosporins alongside an additional anaerobic coverage are the most frequently used agents for PAP in abdominal procedures [4,7,21]. The grouping of cephalosporins are based on the varying in vitro efficacy against gram-positive and/or gram-negative bacteria, with known poor anaerobic activity for all classes of cephalosporins. As a member of 1st generation cephalosporins, CFZ is defined as one of the most efficacious agents against methicillin-sensitive staphylococci with less activity against gram-negative bacteria compared to the ascending order of cephalosporin generations [22]. On the other hand higher generations of cephalosporins are linked to a specific resistance pattern by inducing extended-spectrum ß-lactamase (ESBL) producing gram-negative bacteria [23,24]. In order to harmonize the hospitals’ antibiotic consumption, the prescription culture on surgical antimicrobial prophylaxis was one of the first targets. Subsequently the internal AMS group implemented an in-hospital guideline on PAP directing the agent selection and duration, taking AMS principles and local antibiotic and resistance profile into account. Our amended choice of regimen combines since 2017 a 1st generation cephalosporin with metronidazole (in both analyzed groups either CXM or CFZ was combined with metronidazole, if lower gastrointestinal tract infection was suspected preoperatively and goes along with published data so far that an antimicrobial combination of aerobic and anaerobic coverage is most effective in reducing procedure-associated SSIs in abdominal/colorectal surgery when compared to cephalosporins used as a single agent without having the anaerobic flora of the bowel covered [4,21,25]. The optimal choice however remains still debatable for different groups of cephalosporins have been widley evaluated (with or without metronidazole) to date, confirming three facts: firstly, abdominal surgery without PAP is associated with high SSIs, secondly, the combination of an aerobic agent with an anti-anaerobic antibiotic shows a higher decrease in SSIs and last but not least regarding the choice between first-and second-generation cephalosporins neither demonstrates significantly less SSIs. A Cochrane review on antimicrobial prophylaxis for colorectal surgery checked comparisons between PAP vs. no treatment control/placebo, PAP with additional aerobic coverage vs. same regimen with no additional aerobic coverage, PAP with additional anaerobic coverage vs. same regimen without anerobic coverage, PAP with aerobic coverage vs. only anaerobic regimes and confirmed the first and second statement, whereas comparisons between CFZ and CXM are hardly possible as our study reflects a first when taking a meta-analysis on controlled trials of PAP in biliary tract surgery by Meijer et al. out of equation which by the way revealed no significant differences between CFZ and CXM [21,26]. Randomized controlled trials for SAP in colorectal surgery on CFZ vs. 3rd generation cephalosporins like ceftriaxone or cefotaxime or CFZ vs. cefoxitin, another 2nd generation cephalosporin are available with discrepant results, but without having an aerobic coverage in the analyzed regimens [27,28].

Our findings are in keeping with the rare data published and demonstrate neither a statistically significant higher rate of SSIs, nor a difference in respect to postoperative morbidity or mortality and no LOS when CFZ instead CXM was used for PAP. Nevertheless it must be mentioned that the rate of other postoperative infections such as urinary tract infections or pneumonia were significantly higher in the CFZ group. This trend was also seen by Surat et al. in in cardiothoracic procedures following a switch from CXM to CFZ for surgical prophylaxis [29]. A review by Woodfield et al. demonstrated a reversed effect on both infectious diseases when using single shot ceftriaxone for SAP in abdominal surgery, and the meta-analysis by Dietrich et al. showed a benefit towards urinary tract infections by using ceftriaxone as PAP, but until now no evident statement can be made on the way of influence or differences regarding side-actions by using first- or second generations cephalosporins in abdominal surgery [30,31].

There was no surprise about the full compliance of the surgeons after changing the standard drug (our data show nearly 100% adherence to the new PAP-protocol in terms of drug selection, Table 1). Once the switch of PAP was official, we did expect that this would affect the choice for PAT in indicated cases too (e.g., for peritonitis, Table 1 and Table 4). The European Committee on Antimicrobial Susceptibility Testing (EUCAST) provides break points only for uncomplicated urinary tract infections on selected enterobacterales (e.g., *E. coli* and *Klebsiella* spp.) for cefazolin, leaving no breakpoints for enterobacterales in complicated infections, whereas our data confirm that the majority of the culture findings in an abdominal source constitutes of enterobacterales [7,21,32]. Nevertheless previous published study by Surat et al. suggested that CFZ for the PAT is safe, and this on the other side prompts two questions in patients with achieved source control: does the antibiotic per se make any difference and further, would a single shot even suffice in abdominal infections with e.g., peritonitis [20]. We claim to answer these questions in future trials.

This study has several limitations to be listed: it is monocentric and single-center, it was retrospectively analyzed and the power of the study does not suffice to allow evident conclusions. 

In conclusion, perioperative antibiotic prophylaxis might be safely maintained by CFZ even in the treatment of intra-abdominal infections. Especially, we did not observe a higher incidence of postoperative SSIs after PAP using CFZ.

## Figures and Tables

**Table 1 antibiotics-11-00501-t001:** Change of perioperative antibiotic prophylaxis from cefuroxime to cefazolin during 2016–2019.

	Total	Year				*p* Value
		2016	2017	2018	2019	
Perioperative antibiotic prophylaxis	587 (100.0)	140 (100.0)	134 (100.0)	169 (100.0)	144 (100.0)	1
CFZ	367 (62.5)	0	70 (52.2)	155 (91.7)	142 (98.6)	<0.001
CXM	220 (37.5)	140 (100.0)	64 (47.8)	14 (8.3)	2 (1.4)	
Postoperative antibiotic therapy	168 (28.6)	46 (32.9)	40 (29.9)	45 (26.6)	37 (25.7)	0.520
CFZ	86 (14.6)	0	12 (8.9)	37 (21.9)	37 (27.7)	<0.001
CXM	82 (13.9)	46 (32.9)	28 (20.9)	8 (4.7)	0	

Patients, No. (%); Abbreviations: CFZ, cefazolin; CXM, cefuroxime.

**Table 2 antibiotics-11-00501-t002:** Patient characteristics and intraoperative findings.

Characteristic	Perioperative Antibiotic Prophylaxis		*p* Value
	CXM (*n* = 220)	CFZ (*n* = 367)	
Sex ratio (M:F)	115:105	187:180	0.757
Age, years, mean (SD)	49.8 (21.4)	47.3 (20.0)	0.189
BMI, mean (SD)	26.6 (5.6)	26.8 (6.6)	0.571
ASA ≥ III	51 (23.2)	71 (19.3)	0.338
Comorbidity ^a^, mean (SD)	1.9 (2.3)	1.6 (2.4)	0.072
None (CCI: 0 pts.)	100 (45.5)	188 (51.2)	0.066
Low (CCI: 1–2 pts.)	42 (19.1)	85 (23.2)	
Moderate (CCI: 3–4 pts.)	44 (20.0)	46 (12.5)	
Severe (CCI: >4 pts.)	34 (15.5)	48 (13.1)	
Liver Cirrhosis	0	4 (1.1)	0.121
Chronic Kidney Disease	16 (7.3)	18 (4.9)	0.235
Current Immunosuppressive Drugs	5 (2.3)	16 (4.4)	0.188
Malignant Tumor Disease	16 (7.3)	25 (6.8)	0.832
Preoperative known MDR	2 (0.9)	15 (4.1)	0.026
VRE	0	8 (2.2)	
3MRGN	2 (0.9)	5 (1.4)	
multiple	0	2 (0.5)	
Focus IAIs			<0.001
Acute appendicitis	120 (54.5)	217 (59.1)	
Acute cholecystitis	73 (33.2)	58 (15.8)	
Obstructive Ileus	7 (3.2)	70 (19.1)	
Other ^b^	20 (9.1)	22 (6.0)	
Community-acquired IAIs	203 (92.3)	342 (93.2)	0.677
Hospital-aquired IAIs	17 (7.7)	25 (6.8)	
Intraoperative peritonitis	47 (21.4)	93 (25.3)	0.274
Grade of peritonitis ^c^			0.625
low (MPI ≤ 20)	40 (85.1)	76 (81.7)	
middle (MPI 20–30)	6 (12.8)	15 (16.1)	
high (MPI ≥ 30)	1 (2.1)	2 (2.2)	
Duration of surgery, min, mean (SD)	76.3 (42.6)	73.6 (44.9)	0.125
Laparotomy	40 (18.2)	96 (26.2)	0.031
Laparoscopy	152 (69.1)	233 (63.5)	
Conversion	28 (12.7)	38 (10.4)	
Adequate surgical source control	219 (99.5)	364 (99.2)	0.605
Postoperative transmission to general wards	154 (70.0)	272 (74.1)	0.280

^a^ according to Charlson comorbidity index (CCI); ^b^ e.g., Perforation in upper GI, small intestine, colon; ^c^ according to Mannheimer Peritonitis Index; Abbreviations: CFZ, cefazolin; CXM, cefuroxime; SD, standard deviation; BMI, body mass index; IQR, interquartile range; ASA, American Society of Anesthesiologists; MDR, multi-drug resistant bacteria, at least one antimicrobial drug in three or more antimicrobial categories showed antimicrobial resistance [19]; VRE, Vancomycin-resistent Enterococcus; IAIs, intra-abdominal infections; MPI, Mannheimer Peritonitis Index.

**Table 3 antibiotics-11-00501-t003:** Preoperative patient characteristics and intraoperative findings after propensity score matching.

Characteristic	Perioperative Antibiotic Prophylaxis	
	CXM (*n* = 196)	CFZ (*n* = 196)
Sex ratio (M:F)	98:98	71:125
Age, years, mean (SD)	47.9 (21.2)	48.1 (20.6)
ASA ≥ III	37 (18.9)	35 (17.9)
Preoperative known MDR	2 (1.0)	5 (2.6)
Focus IAIs		
Acute appendicitis	120 (61.2)	118 (60.2)
Acute cholecystitis	55 (28.1)	56 (28.6)
Obstructive Ileus	7 (3.6)	7 (3.6)
Other ^a^	14 (7.1)	15 (7.7)
Community-acquired IAIs	187 (95.4)	188 (95.9)
Hospital-aquired IAIs	9 (4.6)	8 (4.1)
Laparotomy	29 (14.8)	28 (14.3)
Laparoscopy	145 (74.0)	139 (70.9)
Conversion	22 (11.2)	29 (14.8)
Adequate surgical source control	195 (99.5)	194 (99.0)
Postoperative transmission to general wards	149 (76.0)	151 (77.0)

^a^ Perforation in upper GI, small intestine, colon; Abbreviations: CFZ, cefazolin; CXM, cefuroxime; SD, standard deviation; ASA, American Society of Anesthesiologists; IAIs, intra-abdominal infections; MDR, multi-drug resistant bacteria.

**Table 4 antibiotics-11-00501-t004:** Antibiotic therapy and microbiology.

Characteristic	Perioperative Antibiotic Prophylaxis		*p* Value
	CXM (*n* = 196)	CFZ (*n* = 196)	
Repeated intraoperative dose (if duration of surgery ≥ 180 min)	1/5 (20)	5/6 (83.3)	0.036
PAP combination with MTZ	190 (96.9)	187 (95.4)	0.430
PAT	61 (31.1)	58 (29.6)	0.742
Later PAT	3 (1.5)	10 (5.1)	0.207
PAT duration, days, median (IQR)	6 (4–7)	5 (3–7)	0.398
In-house SOP compliance			
PAP	169 (86.2)	171 (87.2)	0.585
Indication for PAT ^a^	51 (83.6)	51 (87.9)	0.500
PAT too long	30 (49.2)	21 (36.2)	0.176
Switch of PAT	26 (13.3)	29 (14.8)	0.746
Intraoperative sampling	96 (49.0)	107 (54.6)	0.102
Positive culture findings ^b^	64 (64.6)	68 (63.6)	0.870
Grampositive bacteria	35 (35.4)	36 (33.6)	0.797
*Enterococcus faecium*	4 (6.3)	6 (8.8)	0.745
*Enterococcus faecalis*	3 (4.7)	1 (1.5)	0.283
*S. aureus*	0 (0.0)	1 (1.5)	0.332
Gramnegative bacteria	49 (49.5)	55 (51.4)	0.785
*E. coli*	35 (54.7)	37 (54.4)	0.975
*Pseudomonas* spp.	7 (10.9)	4 (5.9)	0.355
*Klebsiella* spp.	8 (12.5)	14 (20.6)	0.248
Anaerobic bacteria	30 (30.3)	26 (24.3)	0.333
MDR	2 (1.0)	2 (1.0)	1

^a^ according to AMS; ^b^ pre- or intraoperative sampling; Abbreviations: MTZ, metronidazole; CFZ, cefazolin; CXM, cefuroxime; SOP, standard operating procedure; PAP, perioperative antimicrobial prophylaxis; PAT, postoperative antibiotic therapy; IQR, interquartile range.

**Table 5 antibiotics-11-00501-t005:** Postoperative outcome.

Characteristic	Perioperative Antibiotic Prophylaxis		*p* Value
	CXM (*n* = 196)	CFZ (*n* = 196)	
SSIs	7 (3.6)	14 (7.1)	0.117
superficial	2 (1.0)	6 (3.1)	
deep	3 (1.5)	2 (1.0)	
organ space	2 (1.0)	6 (3.1)	
Other postoperative Infection	4 (2.0)	17 (8.7)	0.004
respiratory	1 (0.5)	4 (2.0)	
catheter	1 (0.5)	2 (1.0)	
urinary	2 (1.0)	7 (3.6)	
other	0 (0.0)	4 (2.0)	
Postoperative Complications ^a^			
none	126 (64.3)	130 (66.3)	0.157
no severe complications ^b^	55 (28.1)	49 (25.0)	
severe complications ^c^	15 (7.7)	17 (8.7)	
Mortality	0 (0.0)	2 (1.0)	0.157
New MDR ^d^	0 (0.0)	2 (1.0)	0.157
LOS, days, median (IQR)	4 (3–7)	4 (3–7)	

^a^ according to Clavien Dindo; ^b^ Clavien Dindo Grade I–IIIa; ^c^ Clavien Dindo Grade IIIb–V; ^d^ within 30 days postoperative; Abbreviations: CFZ, cefazolin; CXM, cefuroxime; SSIs, surgical site infections; PAT, postoperative antibiotic therapy; MDR, multi-drug resistant bacteria; LOS, length of stay; IQR, interquartile range; LOIS, length of stay on ICU.

## Data Availability

The data presented in this study are available on request from the corresponding author. The data are not publicly available due to European General Data Protection Regulation (GDPR).

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
