# Peer review of "Cefazolin Might Be Adequate for Perioperative Antibiotic Prophylaxis in Intra-Abdominal Infections without Sepsis: A Quality Improvement Study"

_antibiotics, 2022, doi:10.3390/antibiotics11040501_

Round 1

Reviewer 1 Report

Thank you for the opportunity to review the study on “Cefazolin is Adequate for Perioperative Antibiotic Prophylaxis in Intra-Abdominal Infections without Sepsis: A quality improvement study”. The study's purpose of the study was to compare the risk of SSIs between the second-generation cefuroxime (CXM) and the first-generation cefazolin (CFZ) used as perioperative antibiotic prophylaxis in intra-abdominal infections (IAIs) without sepsis. Overall the manuscript was well written and the study was well conducted. However, a few issues need to be addressed in their current form.

These are the comments for the authors:

Title

The title did not reflect the objective. It reflects as a clinical trial instead of a quality improvement study. Please do kindly revise accordingly

Abstract

The abstract and methodology were presented as a non-inferior study design and did not reflect the quality improvement aspect.

Introduction

It was adequate.

Methodology

  1. Please do justify or explained how the non-inferior margin of 8% was determined?
  2. The non-inferior study is best to be presented as relative difference and absolute difference with 95%CI.
  3. In the discussion, the authors mention ‘in both analyzed groups either CFX or CFZ was combined with metronidazole, although depending on the type of procedure metronidazole may not be necessary’. This statement should place in the limitation segment. The authors are suggested to include metronidazole in the analysis to ensure that the findings of this study were not confounded by the use of metronidazole.

Results

  1. The results are well presented and written. However, the manuscript's terminology ‘perioperative antibiotic therapy’ is rather confusing. Is the author trying to indicate perioperative prophylaxis (PAP) or postoperative antibiotic treatment (PAT)? Please do revise accordingly.
  2. The mean timing of the first dose of PAP administration was not provided.
  3. Did the study exclude intraoperative addition dose of PAP since this was a study on perioperative antibiotic prophylaxis? If the study includes this data, it is good to present it in table 3.
  4. The results showed that there was a significant difference in ‘other postoperative infections’ between both groups. It is hardly justified the conclusion that the CFZ is non-inferior to CMX merely based on SSI alone.
  5. The result for segment 3.5 was difficult to understand. The phrase ‘Interestingly, the incidence of SSIs was higher in patients requiring PAT (8.2% with CXM, 13.8% with CFZ) in comparison to single-shot PAP (1.5% with CXM, 4.3% with CFZ)’ was uncleared. What was the author trying to highlight? It is hard to justify the conclusion as mentioned in the abstract.
  6. For tables 4& 5, the table heading ‘perioperative antibiotic therapy’ was confusing. Suggest to revise

Discussion

  1. More statistical evidence is needed to support the discussion.

Conclusion

The conclusion of the study is not available.

Some minor comments

  1. A minor typo in the discussion segment, page 9 second paragraph ‘Once the swich was official we…’
  2. A minor typo in the discussion segment ‘in both analyzed groups either CFX or CFZ was combined with metronidazole, although depending on the type of procedure metronidazole may not be necessary’

Author Response

Dear Reviewer

Many thanks for your comprehensive revision of our article. Please find the corrections below:

1) Title: We did modify the title accordingly. We hope this will suffice.

2) Abstract: This was amended as wished. Thanks.

3) 1/2: Sample seize calculation was done in order to determine the power of the study. SSIs occur in up to 10% of colorectal surgery, therfore we choose 8% as an acceptable upper margin to presume non-inferiority of cefazolin.

3) 3: We modified the misleading sentence concerning the addition of metronidazole.

Results: 1) Thank you for your remark. We revised that aspect.

2) We are happy to add that the administration of PAP was given in accordance to national/international guidelines within 30-60 minutes prior skin incision, but we  are not able to provide the individual timings.

3) This was added to table 4.

4) Thank you, I understand your notion. The main aspect of this study concerned SSIs though, the reasons for higher postoperative infections due only to having changed the drug for PAP would be presumptuous. PAP is primarily given to reduce SSIs, its use is not intended to reduce other infections than SSIs patients might develop during their postoperative stay, so every positive/negative effect remains a side-action of unknown rationale. But I must admit one that has to be subject of further investigations.

5) The wording was adapted. 

6) Thank you, has been revised.

Discussion:

We tried to modify the discussion, but it is not easy to balance the requirements between all reviewers.

Minor changes admended. Thanks again.

Sincerely

Güzin Surat

Reviewer 2 Report

I read with interest the manuscript by Surat et al and I have following concerns:

1. The aim of the study is not clearly described and a summary of main findings not mentioned in the first paragraph of discussion.

2. Please describe in more detail the Clavien-Dindo severity classification system.

3.In methods the authors write “A structured local PAP guideline was implemented in May 2017 and set CXM instead CFZ as the new standard prophylactic antibiotic”. In results the authors write “Until May 2017 CXM had been the local standard agent for PAP (in combination with the anti-anaerobic antibiotic metronidazole). Equally 100% of patients in 2016 received CXM, not one patient CFZ. Vice versa the amount of patients receiving CFZ for PAP reached then nearly 100% in 2019.” It’s a bit confusing which antibiotic was used first and which was implemented later on. Or the paragraph in Results section describes the low adherence to the ASP? Please clarify.

4. It is not clear the population investigated. All patients suffered by an intraabdominal infection but not sepsis? Elective surgery was excluded? Please clarify? I understand that prophylaxis is administered for elective surgery but in case of IAI is it sure that only 1 dose of prophylaxis was given before surgery and not a longer course of therapy?

5. How adequate source control (pg 4) is defined?

6.How MDR is defined?

7. Comorbidity low, moderate, severe is not described in the paper by Charlson. Please provide mean or median value for each group and SD or IQR and compare accordingly.

8. Subtitle “Antibiotic therapy and microbial findings” Please replace “microbial” with a more appropriate term (eg microbiology). Same for resistogram. In general, English use requires attention and should profit from native speaker editing

9. Nothing is mentioned about Ethics Committee approval of the study.

10. Please revise Discussion section to be shorter and more to-the-point.

Author Response

Dear Reviewer

Many thanks for your feedback. Please find our comments below:

1) Thanks, quite right. We modified that first part accordingly.

2) The classification system of Clavien-Dindo is a quite commonly used in surgery, and has been applied in many studies before, therefore we would like to leave as it is.

3) We are truly sorry, the confusion is clarified now.

4) All patients were diagnosed with intra-abdominal infections, and underwent emergency surgery, none of the patients were septic. We tried to mention this as often as possible.

5) + 6) Definitions added.

7) Has been amended.

8) Thanks, we agree and did amend that point.

9) It comes up at the end of the article.

10) Thanks very much, but it is not always easy to balance the requirements between each reviewer, but we tried to modify the discussion part.

Reviewer 3 Report

In the Abstract you have Conclusion: Perioperative antibiotic prophylaxis can be safely maintained by CFZ even in the treatment of intra-abdominal infections.

Please make a Conclusion at the end of the text.

What you have stated (This study has several limitations to be listed: it is monocentric and single-centered, it was retrospectively analyzed and the power of the study does not suffice to allow evident conclusions), can be neither a conclusion nor a conclusion .

Author Response

Thank you very much, dear reviewer.

Has been amended accordingly.

BW

Güzin Surat

Round 2

Reviewer 2 Report

The authors have addressed all my concerns.

This manuscript is a resubmission of an earlier submission. The following is a list of the peer review reports and author responses from that submission.